# Depth of Invasion: Influence of the Latest TNM Classification on the Prognosis of Clinical Early Stages of Oral Tongue Squamous Cell Carcinoma and Its Association with Other Histological Risk Factors

**DOI:** 10.3390/cancers15194882

**Published:** 2023-10-08

**Authors:** Ignacio Navarro Cuéllar, Samuel Espías Alonso, Francisco Alijo Serrano, Isabel Herrera Herrera, José Javier Zamorano León, José Luis Del Castillo Pardo de Vera, Ana María López López, Cristina Maza Muela, Gema Arenas de Frutos, Santiago Ochandiano Caicoya, Manuel Tousidonis Rial, Alba García Sevilla, Raúl Antúnez-Conde, José Luis Cebrián Carretero, María Isabel García-Hidalgo Alonso, José Ignacio Salmerón Escobar, Miguel Burgueño García, Carlos Navarro Vila, Carlos Navarro Cuéllar

**Affiliations:** 1Oral and Maxillofacial Surgery Department, Hospital General Universitario Gregorio Marañón, 28007 Madrid, Spain; nnavcu@hotmail.com (I.N.C.); dra.amlopezlopez@gmail.com (A.M.L.L.); crismazamuela@gmail.com (C.M.M.); gema.arenas@gmail.com (G.A.d.F.); sochandiano@hotmail.com (S.O.C.); manuel@tousidonisrial.com (M.T.R.); albagsevilla@gmail.com (A.G.S.); jisalmeron@telefonica.net (J.I.S.E.); cnavarrovila@gmail.com (C.N.V.); cnavarrocuellar@gmail.com (C.N.C.); 2Private Practice, Sta. Susana, 41, 33007 Oviedo, Spain; saespias@gmail.com; 3Pathology Department, Hospital General Universitario Gregorio Marañón, 28007 Madrid, Spain; pacoalijo@hotmail.com; 4Radiology Department, Hospital General Universitario Gregorio Marañón, 28007 Madrid, Spain; isabel.herherrera@gmail.com; 5Public Health and Maternal & Child Health Department, School of Medicine, Universidad Complutense, 28040 Madrid, Spain; jjzamorano@ucm.es; 6Oral and Maxillofacial Surgery Department, Hospital Universitario La Paz, 28046 Madrid, Spain; rodrigator2001@hotmail.com (J.L.C.C.); mburguenom@me.com (M.B.G.); 7Oral and Maxillofacial Surgery Department, Hospital Universitario Ruber Juan Bravo, 28006 Madrid, Spain; antunezconde_92@hotmail.com; 8Radiology Department, Hospital Universitario Puerta de Hierro, 28222 Madrid, Spain; mabelgha@gmail.com

**Keywords:** tumour depth of invasion, squamous cell carcinoma of the head and neck, oral cancer, TNM classification, lymph node metastases, perineural invasion, tumour differentiation grade

## Abstract

**Simple Summary:**

Tumour depth of invasion is a well-known histological risk factor in oral cavity squamous cell carcinoma. With the advent of the eighth edition of the American Joint Committee on Cancer in 2017, a consensus has been established on the point from which to measure the depth of tumour invasion. All of this has led to changes in the T category of the TNM classification, leading to an increase in staging and possible adjustments in the management of adjuvant treatments. The main objective of this research is to assess the prognosis, according to the depth of invasion in patients with clinical early stages of oral tongue squamous cell carcinoma, to evaluate the influence of the depth of invasion in the latest TNM classification as well as in the global staging system, and to analyse its relation to other histological risk factors.

**Abstract:**

Background: The American Joint Committee on Cancer (AJCC), in its 8th edition, introduces modifications to the previous TNM classification, incorporating tumour depth of invasion (DOI). The aim of this research is to analyse the prognosis (in terms of disease-free survival and overall survival) of clinical early stage (I and II) squamous cell carcinomas of the oral tongue according to the DOI levels established by the AJCC in its latest TNM classification to assess changes to the T category and global staging system and to evaluate the association between DOI and other histological risk factors. Methods: A retrospective longitudinal observational study of a series of cases was designed. All patients were treated with upfront surgery at our institution between 2010 and 2019. The variables of interest were defined and classified into four groups: demographic, clinical, histological and evolutive control. Univariate and multivariate analyses were carried out and survival functions were calculated using the Kaplan–Meier method. Statistical significance was established for *p* values below 0.05. Results: Sixty-one patients were included. The average follow-up time was 47.42 months. Fifteen patients presented a loco-regional relapse (24.59%) and five developed distant disease (8.19%). Twelve patients died (19.67%). Statistically significant differences were observed, with respect to disease-free survival (*p* = 0.043), but not with respect to overall survival (*p* = 0.139). A total of 49.1% of the sample upstaged their T category and 29.5% underwent modifications of their global stage. The analysis of the relationship between DOI with other histological variables showed a significant association with the presence of pathological cervical nodes (*p* = 0.012), perineural invasion (*p* = 0.004) and tumour differentiation grade (*p* = 0.034). Multivariate analysis showed association between depth of invasion and perineural invasion. Conclusions: Depth of invasion is a histological risk factor in early clinical stages of oral tongue squamous cell carcinoma. Depth of invasion impacts negatively on patient prognosis, is capable per se of modifying the T category and the global tumour staging, and is associated with the presence of cervical metastatic disease, perineural invasion and tumoural differentiation grade.

## 1. Introduction

Lip and oral cavity carcinomas account for 2% of the incidence and 1.8% of cancer mortality worldwide [1]. Oral cavity squamous cell carcinoma (SCC), including all ethnic groups, is generally 2–3 times more frequent in male patients than in female patients, with the tongue being the most common location (40% of the patients) [2]. Oral tongue cancer represents 0.93% of newly diagnosed cases and 0.45% of cancer-related deaths in the United States [3].

The initial clinical stages (I and II) of SCC of the tongue are represented by those tumours classified as T1 or T2 and N0. In the treatment of this disease, there is a large consensus that the tumour should be extensively excised with wide margins [4] while the neck approach can be performed in different ways, including surveillance [5], elective dissection [6,7], selective sentinel lymph node biopsy [8] and elective radiotherapy [9].

Tumour depth of invasion is a widely documented and reported histological risk factor in oral cavity carcinomas, so the deeper a tumour is, the more likely it is that the disease will spread to the lymph nodes in the neck region [10,11,12,13,14,15,16,17,18]. Lydiatt et al. [19], in 2017, report a distinction between the terms “depth of invasion” and “tumour thickness”, whereby depth of invasion (DOI) refers to the distance (measured in mm) between the level of the basement membrane relative to the closest intact squamous mucosa and the deepest point of invasion of the carcinoma, while tumour thickness is defined as the distance (measured in mm) between the outermost point of the tumour and the deepest point of invasion of the neoplasm.

The American Joint Committee on Cancer (AJCC), in its eighth edition and subsequent update, introduces modifications with respect to the previous TNM classification, based on the scientific evidence described in the literature [19,20]. Essentially, the two concepts incorporated are depth of invasion (DOI) and extracapsular or extranodal extension of the tumour in the lymph nodes (ENE). Thus, both parameters can modify the “T” and “N” categories, respectively, resulting in tumoural upstaging and potential variations in the management of adjuvant treatments.

The starting point of the research project is based on tumour depth of invasion.

The working hypothesis is as follows: “Depth of invasion, divided into 3 groups according to the 8th edition AJCC classification, worsens the prognosis of patients with initial clinical stages of oral tongue squamous cell carcinoma”.

The main aim is to determine the prognosis of early stages (I and II) oral tongue SCC according to DOI levels established by the AJCC in its latest TNM classification.

Secondary objectives are to analyse the clinical-pathological discrepancy of the T category according to DOI, to describe how DOI influences changes in tumoural staging system within the 7th and 8th editions of the AJCC and to assess the association between DOI and the presence of other pathological risk factors such as positive cervical lymph nodes, perineural invasion (PNI), lymphovascular invasion and tumoural differentiation grade.

## 2. Materials and Methods

A retrospective longitudinal observational study of a series of cases was designed, with the population of people who attended the Oral and Maxillofacial Surgery Department of the Gregorio Marañón University Hospital, as the framework for analysis. This research study is endorsed by the ethics committee of the institution. The time period of this study was between January 2010 and December 2019. The variables of interest were defined by grouping them into 4 groups: demographic, clinical, histological and evolutive control. Information on these variables is collected from the archives of the surgical reports of the Oral and Maxillofacial Surgery Department, the clinical patients files and the archives of histological samples of the hospital’s Pathology Department.

Patients who were enrolled in this study had to meet all of the following inclusion criteria:-Patients had initial clinical stages of oral tongue SCC (cT1, cT2, cN0, stages I, II) according to the eighth edition of the AJCC.-Patients had undergone surgery at the Oral and Maxillofacial Surgery Department of the Gregorio Marañón University Hospital.-Patients’ surgical procedures consisted of extensive tumour removal and elective neck dissection.-The tumour margins of the surgical resection were negative.-Patients were over the age of 18.

To be included in the research, NONE of the following criteria could be met:
-Disease that extended beyond the anterior two thirds of the tongue, therefore affecting the oropharynx.-Patients, in whose medical records, variables of interest were not identified.-Patients who were previously treated for carcinoma of the oral cavity (surgery, radiotherapy, or chemotherapy).

After establishing the inclusion and exclusion criteria and reviewing the radiological tests and histological data, a final sample size of 61 patients was obtained (Figure 1).

A descriptive analysis of the different variables grouped into demographic, clinical, histological and evolutive control was performed. The results were presented as the frequency and percentage of each analysis group for qualitative variables and as mean ± standard deviation of the mean for quantitative variables. Normality distribution analysis was performed using the Kolmogorov–Smirnov test. The analysis of frequency distribution between the different groups of interest was carried out using chi-squared (χ^2^) and Fisher’s exact test. To analyse the influence of the histopathological marker DOI on overall survival, disease-free survival and recurrences (measured from the date of surgery to the occurrence of events), survival analyses were performed using Kaplan–Meier (KM) survival functions, as well as the graphical representation of the KM curves. The difference between pathological T stages according to DOI was determined using the log-rank test of the curves. In the analytical statistical phase, we worked with a two-tailed type-1 error, establishing statistical significance for *p*-values lower than 0.05. Statistical analysis was performed with SPSS software version 25.0.

## 3. Results

### 3.1. Total Sample Descriptive Analysis

Table 1, Table 2, Table 3 and Table 4 show the demographic, clinical, histological and evolutive control data of the total research sample (*n* = 61).

In general (Table 1), the profile of the patients studied is mainly male, of approximately 60 years of age, with no associated risk factors (tobacco and alcohol consumption). Only 18% of the population reported drinking, while the percentage of smokers was close to 43%.

From a clinical point of view (Table 2), it is noteworthy that the distribution of tumours according to size is practically symmetrical between T1 and T2 groups, with the lingual lateral border being the most frequent site of lesion presentation (82%).

All but one of the patients were studied with Computed Tomography scan (CT scan) and only in the 23% of them a Magnetic Resonance (MR) was obtained, which is probably related to the celerity with which CT scan versus MR is acquired. Interestingly, the radiological size of the tumour could not be assessed in approximately 39% due to the presence of amalgams and dental artefacts or the superficiality of certain tumours.

Almost half of the patients were reconstructed in a simple way with direct closure. Whether unilateral or bilateral neck dissection was performed depended on the location of the tumour. More than half of the sample underwent adjuvant radiotherapy and only two patients were treated with chemotherapy.

A total of 72% of patients had a moderate tumour differentiation grade, with PNI present in 40% of the sample and lymphovascular invasion in only five patients (Table 3). About 34% of patients had pathological lymph neck nodes, with extracapsular involvement being anecdotal (only three patients).

A relevant fact is the difference in the T category between the pathological T in the seventh and eighth editions of the AJCC. As can be seen in Table 3, according to the seventh edition, only 5% of patients is pT3, while in the eighth edition, this increases to 34%. This is explained by the importance given to DOI in the eighth edition.

In the histological neck variable, there is no N2a category in the seventh edition as there were no patients with lymph nodes between 3 and 6 cm in size. However, one patient was included in this category in the eighth edition because he had a pathological lymph node with extracapsular involvement (ENE+) and a size of less than 3 cm. The pN3b category is also not represented in the seventh edition, while two patients can be included in the eighth edition because they have more than one positive cervical lymph node with at least one of them having ENE.

In the staging variable according to the seventh edition, there are no categories of IVb and IVc, as there were no patients with N3 or metastatic disease (M1). However, according to the eighth edition, category IVb is included as there were two pN3b patients.

It is worth noting (Table 4) that loco-regional and total recurrence have the same values (23% of the sample) since the five patients who presented distant recurrence also did so at a loco-regional level. Twelve patients died (19.67%), of which eight were due to the disease, two to carcinomas in other locations, one to pneumonia and another whose cause of death was not recorded in the clinical history.

### 3.2. Descriptive Analysis of the Sample Divided into Subgroups Depending on T Pathological According to DOI

Table 5, Table 6, Table 7 and Table 8 outline the demographic, clinical, histologic and evolutive control characteristics of the total sample population divided into three subgroups:-Subgroup pT1 according to DOI (*n* = 17): DOI ≤ 5 mm.-Subgroup pT2 according to DOI (*n* = 24): DOI between 5 and 10 mm.-Subgroup pT3 according to DOI (*n* = 20): DOI > 10 mm.

When the demographic characteristics of the sample were analysed (Table 5), no significant differences were found in the distribution between the categories of the variables analysed in the different subgroups, depending on pathological T according to DOI.

Statistically significant differences were found in the distribution between pathological T according to DOI (divided into three subgroups) and the different categories of the following clinical variables (Table 6): clinical tumour size, radiological tumour size, elective neck dissection, type of reconstruction and postoperative treatment with radiotherapy.

Concerning tumour size, about 50% of the tumours categorised as T2 became pT3 when DOI was analysed, while this percentage was only about 15% in the T1 category. Overall, about 50% of patients increased their pT when considering DOI, highlighting its importance for correctly classifying the T category.

Regarding the type of elective neck dissection, of the 28 patients who received bilateral lymphadenectomy, 82% were clinically T2, and it is remarkable that 60% of the patients who received a bilateral dissection were classified into the pT3 group, while only 9% of those who underwent unilateral lymphadenectomy were designated to this same subgroup.

A total of 50% of the patients reconstructed with direct closure were subclassified in the pT1 group while 72% of microsurgical techniques were employed in pT3 patients.

It is very relevant that 62% of the patients who received postoperative radiotherapy (PORT) were pT3 while about 44% of those who were not irradiated were classified as pT1.

When analysing histological variables (Table 7), statistically significant differences were found in the distribution between pathological T according to DOI (divided into three subgroups) and the different categories of the following variables: DOI, tumour differentiation grade, PNI, pathological T according to size, pathological T of the 7th and 8th edition of the AJCC, histological neck, histological positive neck and the staging according to the 7th and 8th edition of the AJCC.

About 66% of the well-differentiated lesions were observed in the pT1 subgroup, while the vast majority of moderately (77%) and poorly (100%) differentiated carcinomas were associated with the pT2 and pT3 subgroups, suggesting that the more undifferentiated tumours were associated with a depth greater than 5 mm.

Out of the 40% of patients with perineural invasion, more than 50% were associated with pT3 tumours, showing that the higher the DOI, the more likely it is to be associated with PNI.

A total of 50% of pT1 patients by tumour size were also pT1 by DOI, while more than half of the patients with a pT2 neoplasm by size were upstaged to pT3 when DOI was measured. Thus, there is a considerable increase in the T category when DOI is measured (50% in pT1 and about 54% in pT2 measured by size).

When analysing the overall pathological T according to the 7th and 8th editions of the AJCC, it can be seen that according to the 7th edition (only measured by size), there would be three patients (5%) with pT3 tumours; whereas, when taking into account the 8th edition (measured by size and DOI), a total of 21 patients would be reclassified as pT3.

Approximately one third of the sample analysed had lymph node metastases. Of the 21 patients with positive necks, 14% corresponded to the pT1 subgroup and 57% to pT3, with a higher probability of lymph node involvement associated with deeper tumours. Four patients developed diseases in both side of the necks and three of them were in the pT3 subgroup (DOI > 10 mm).

When the staging systems of the seventh and eighth editions of the AJCC are evaluated, it can be appreciated that in the early stages I and II (T1N0 and T2N0), DOI is very relevant. The percentages of patients in stages I and II, according to the seventh edition, are 41% and 24%, respectively, while in the eighth edition, they are 21% and 31%. The decrease in patients in pT1 and the increase in pT2 can only be explained by tumoural DOI, as the neck is negative (pN0) in both stages. There are three patients with ENE, which implies that, according to the 8th edition, they should be classified as pN2a and pN3 and, therefore, upgrade to stage Iva (one patient) and Ivb (two patients). According to the 7th edition, no patient would be stage IV as it does not consider ENE as a staging criterion.

Concerning the analysis of the evolutive follow-up variables (Table 8), no significant differences were observed in the distribution between the different categories of the variables analysed in the three subgroups, depending on pathological T according to DOI.

Despite the marked differences in total recurrence between groups pT1 and pT3, statistical significance was not found. This is probably related to the limited sample size. In terms of mortality, half of the deaths occurred in the pT3 group.

The regression used for the multivariate analysis of all variables that were significant revealed a crucial association between the DOI and PNI (*p* = 0.039).

### 3.3. Analysis of Primary and Secondary Objectives

#### 3.3.1. Main Aim (DFS, OS)

The mean follow-up time of the study population was 47.42 months.

An analysis of disease-free survival reveals (Figure 2) that 95% of patients assigned to the pT1 subgroup did not relapse during the time of follow-up, while this percentage decreased to 60% in the pT3 subgroup.

The Kaplan–Meier recurrence function curve (Figure 3) is very representative of these data, showing the differences in DFS between the three subgroups, mainly between pT1 and pT3.

Table 9 indicates that there are significant differences in DFS, according to pathological T subgroups, according to DOI, such that the mean time to relapse in pT3 is almost half as long as in pT1 (57.9 versus 108.6 months).

Analysis of overall survival reveals (Figure 4) that 88% of pT1 patients survive while this percentage decreases to 70% in pT3 patients.

The Kaplan–Meier overall survival function curve (Figure 5) shows very little difference between the pT1 and pT2 subgroups and slightly more with pT3.

The mean time to the occurrence of deaths is depicted in Table 10. Although there is a large difference between groups, the required level of significance is not reached. Therefore, the results reveal that there are no significant differences in overall survival according to pathological T according to DOI.

#### 3.3.2. Secondary Objectives

##### Clinical and Pathological Discrepancy of the T Category According to DOI

When analysing the discrepancy between clinical tumour size (clinical T) and pathological T according to DOI, a total of thirty-four patients (55.73%) changed their “T” category, of which thirty where upstaged and four downstaged (four patients with clinical T2 were finally pT1 according to DOI). Considering the overall TNM classification for the T category (size and DOI) of the eighth edition of the AJCC, finally, 32 patients modified their “T” category (*p* < 0.001). Among these thirty-two people, two were downgraded to T1 and thirty were upgraded to either T2 or T3. In summary, 30 patients (49.2%) in the sample increased their T category as outlined in Table 11.

##### How DOI Impacts the Staging System—Comparison of the 7th and 8th Editions of the AJCC

A total of 18 patients changed staging, representing 29.5% of the sample analysed. Of these, most changes were related to the “T” category and only 3 to the “N” category. The reason for this is that the overall staging of these tumours is strongly influenced by neck positivity (pN), which means that, although there are possible changes in the T category caused by DOI, these may not be reflected in the final staging due to the N category. For example, a patient with a 4 mm DOI lesion (pT1) and a positive lymph node (N1) would have the same tumour stage as another with a 12 mm DOI lesion (pT3) and a positive lymph node (N1). Both lesions would be stage III.

##### Correlation of DOI with the Presence of Positive Neck Nodes

The analysis of the results reveals that there are significant differences (*p* = 0.012) between the presence of positive lymph nodes and the subgroups of pathological T according to DOI, such that the deeper the tumour (pT3 group), the greater the probability of presenting neck disease. Figure 6 presents this result very well, showing how the presence of pathological nodes progressively doubles as the pT population subgroup increases.

##### Correlation of DOI with Perineural Invasion

The relation of PNI to pathological T according to DOI (Figure 7) shows statistically significant results (*p* = 0.004) such that patients with tumours (pT1) have a significantly lower incidence of PNI than those with more invasive lesions (pT3).

##### Correlation of DOI with Lymphovascular Invasion

The analysis of results of the relation between lymphovascular infiltration and pathological T according to DOI divided by subgroups does not reach statistical significance (*p* = 0.334). This is probably due to the fact that only five patients presented this histological risk factor.

##### Correlation of DOI with Tumour Differentiation Grade

The results of the analysis of the relation between the tumour differentiation grade (DG) and pathological T according to DOI (Figure 8) divided by subgroups reveal that there is a significant association (*p* = 0.034) between both variables in such a way that there are few patients with well-differentiated tumours in the pT3 subgroup and no patients with a poor differentiated tumour in the pT1 subgroup. Most of the tumours were moderately differentiated.

## 4. Discussion

The treatment options for patients with early stages of oral cavity squamous cell carcinoma (I, II) are focused on the management of the primary tumour and the N0 neck. For the primary neoplasm, the two treatment options are surgery (extensive tumour resection and immediate reconstruction) [4,21,22,23,24] and radiotherapy [25,26]. The management of the N0 neck is more controversial. Valid alternatives are observation and close radiological ultraosund monitoring [5,27], irradiation [9] or surgical management, either by elective neck dissection [6,7] or selective sentinel node biopsy [8,28,29]. All patients in this research underwent tumour excision and elective neck dissection. It was decided to exclude patients with positive tumour margins because of the impact these could have on the prognosis of the disease.

Depth of invasion, as a histological risk factor, has been extensively studied in different tumours of the organism such as melanoma, breast cancer or cervical carcinoma [20]. Moore, in 1986 [30], investigated tumour thickness as a prognostic factor in upper aerodigestive tract carcinomas, emphasising the difference between tumour thickness and depth of invasion and stating that DOI refers to the extent of the cancer below an epithelial surface (basement membrane). Ambrosch et al. [31] illustrate in their research how they measure DOI from a virtual line drawn at the level of the normal mucosa to the point of maximum tumour depth.

Until recently, there was no clear consensus on how to measure DOI. Some authors measure it according to the structures that infiltrate the tumour in depth [32,33], others use different reference points from which to measure this parameter [14,34,35], certain papers do not specify how the authors estimate DOI [36,37], and in many other publications, the concepts of tumour thickness and depth of invasion are considered similar [38,39]. A paper by Giacomarra et al. [40] reports that tumour thickness and DOI are commonly used as synonyms referring to the part of the tumour below the basement membrane. One of the possible reasons for finding such a disparity of criteria is the subjectivity with which the deepest tumour cell can be measured [14].

With the advent of the latest AJCC classification [19,20], criteria have been unified in such a way that the point from which DOI should be measured is the basement membrane. As a result, the different depth measurements have been grouped together and the concept of DOI has been incorporated into the TNM classification.

Among all the parameters that conform the TNM classification, the maximum surface diameter of tumours (tumour size or clinical T) is the most controversial of all for several reasons [41]: tumour size must be accurate, which is not always easy to measure in the oral cavity; prognosis is not necessarily related to the maximum tumour size as there are patients with large lesions who have a very good outcome and others with smaller neoplasms whose survival is poorer; applying 2 cm measurement ranges (e.g., T2 includes 2–4 cm lesions) is too large to assess adjuvant treatments so that patients can be under-treated or over-treated; some tumours are multicentric, which makes measurement more difficult.

The inclusion of DOI and the modification of the variable “primary tumour or T” in the TNM classification overcomes a part of this problem by providing value to the fact that a deep tumour can be more aggressive than a superficial lesion regardless of its size in the mucosa. The starting point of this research is the definition of DOI and its grouping into three subcategories, according to the latest AJCC classification, in such a way that those neoplasms with a tumour infiltration of less than or equal to 5 mm are considered pT1, those between 5 and 10 mm are pT2 and those with a tumour infiltration larger than 10 mm are grouped in pT3. Therefore, DOI, per se, can modify the T category of the TNM classification of oral cavity SCC.

The main aim of this study is to analyse the prognosis in terms of disease-free survival and overall survival of patients with early clinical stages (I, II) of SCC of the oral tongue, according to the depth of tumour invasion indicated by the TNM classification of the eighth edition of the AJCC. Ebrahimi et al. [42] conducted a multicentre study of 3149 patients, prior to the staging modification of the 8th edition of the AJCC. They propose to include DOI in the “T” category” of the TNM classification for those tumours arising in the oral cavity and analyse how the staging would vary, depending only on DOI, with respect to the 7th edition. They report that the increase in DOI was related to an increase in pT and pN categories, to extranodal extension, to positive margins and to more advanced stages of disease. They point out that disease-specific mortality is the worst, with statistically significant results, when DOI is >5 mm in pT1 (<2 cm) and when it is >10 mm in pT2 and pT3 (>4 cm). These authors propose up to five different models of how DOI could become a part of the TNM classification, with 5 and 10 mm being the reference measurements for this purpose. Pollaers et al. [43] report significant differences in the assessment of DFS when comparing the groups pT1 vs. pT2, pT1 vs. pT3 and pT2 vs. pT3. Tirelli et al. [44] identify statistically significant differences between the pT variable according to DOI and DFS, considering DOI to be a poor prognostic factor. Matos et al. [45] carry out a study on 298 patients with oral cavity SCC, concluding that with the inclusion of DOI in the TNM classification, the staging of patients with this disease is better and the OS and DFS worsen. Amit et al. [46] analysed the prognosis, in terms of OS and DFS, of 244 individuals diagnosed with tongue SCC according to the new staging system, observing a worsening of survival parameters between pT2 and pT3. Vuity et al. [47] state that DOI is an independent predictor of OS, together with gender and lymphatic spread, and of DFS together with gender, perineural invasion and neck positivity. Caldeira et al. report [48] a meta-analysis in which they analyse the relevance of DOI and its prognosis in the early stages of oral cavity SCC. After an initial analysis of 1300 publications, only 19 (2404 patients) met the inclusion/exclusion criteria. They conclude that DOI is a good prognosticator for early stage oral cavity SCC because it contributes to a higher recurrence rate and a decreased survival. The findings of the meta-analysis highlight the clinical relevance of DOI and corroborate its incorporation for staging OSCC. Wunschel et al. [49] evaluate the relevance of DOI in 135 patients with oral cavity SCC concluding that depth of invasion is the strongest histologic predictor of metastatic tumour growth, overall survival and relapse-free survival in oral cavity SCC, confirming the current adaption of the T-classification.

The sample analysed in our research reveals a significant association of the three DOI categories (grouped according to the AJCC 8th edition TNM classification) and disease-free survival, but not with overall survival. The fact that overall survival is not influenced by DOI is probably due to the sample size.

In terms of changes in the T category and global staging, when comparing the seventh and eighth editions of the AJCC, different papers report modifications in the T category, ranging from 32.5% to 50%, and in global staging, ranging from 30% to 37% [43,44,45,46]. Matavelli et al. [50] evaluate the improvement in the prognostic power of the 8th AJCC edition compared to the 7th edition. A total of 41% of the patients were upstaged with a 59% of change in the T category and a 49% of staging modification. However, there are other surveys, such as those by Dirven et al. [51], which conclude that the prognostic performance of the T category and TNM stage of the 8th edition of the AJCC staging of oral cancer is similar regardless of whether DOI or thickness is used as a modifier of the T category. In the sample studied in this research work, there was an increase in the T category of 49.2% and a global staging adjustment of 29.5%. This difference is due to the fact that the N category tends to equalise global staging.

The association of DOI with the presence of positive neck nodes is widely described in the scientific literature. [10,11,12,13,14,15,16,17,18]. Faisal et al. [12] analyse the influence of DOI, according to the subgroups of the latest AJCC classification, on local recurrence and on the presence of pathological lymphadenopathies in the early clinical stages of tongue SCC. Among a sample of 167 patients, they found a rate of occult cervical metastases of 36%, 29% for T1 tumours and 45% for T2 neoplasms. When distributing both groups according to DOI, they found that when DOI was ≤ 5 mm (pT1), the percentage of positive necks was 23%; when it was located between 6 and 10 mm (pT2), it rose to 34%; while when it was beyond 10 mm (pT3), it was 53%. Both DOI and the presence of positive lymph nodes were statistically related to local recurrence. Patients with pT3 tumours (DOI > 10 mm) presented worse OS. Our results revealed that cervical lymph node involvement (21 patients, 34.4%) is related to the depth of tumour invasion in such a way that the percentage of patients with positive necks was higher as DOI increased. Thus, 17.64% of pT1 subjects, 25% of pT2 patients and 60% of pT3 patients developed metastatic disease in the neck, highlighting the importance of this histological risk factor in the evolution of the disease.

PNI is the capability of certain tumour cells to invade nerve structures present in the same or nearby tissue where the carcinoma proliferates. It constitutes a pathway for tumour dissemination since the nerve provides a route for the spread of the neoplasm. Some authors [52,53] evaluate IPN as an independent risk factor for the presence of occult cervical metastases and recommend elective neck dissection, depending on whether this histological parameter is present. Rahima et al. [54], on a sample of 101 patients, study the relevance of PNI in oral cavity and oropharyngeal SCC, observing a significant association between PNI and DOI and concluding that a five-year DFS was significantly worse for patients with PNI. Tai et al. [55] report that when DOI is greater than 6 mm, it is associated with PNI, the presence of positive nodes in the neck is higher and DFS is significantly lower. Our research indicates that PNI is significantly associated with the three subgroups of pathological T according to DOI in such a way that the deeper the tumour, the more PNI is identified. Multivariate analysis revealed a worse prognosis in patients with PNI.

Lymphovascular infiltration is defined as the presence of tumour cells in the walls or lumen of blood or lymphatic vessels in the tissue where the primary tumour is located. There are papers [56,57] that state that lymphovascular infiltration is associated with a worsened prognosis of oral cavity SCC, although some of them take this statement with caution [56]. Similarly, there are other research studies that find no statistical association between lymphovascular infiltration and DOI [58]. In our series, the presence of lymphovascular infiltration was low, being found in only five patients, with no statistically significant association between this histological parameter and the three pathological T subgroups according to DOI.

The histological differentiation grade of a tumour refers to the similarity between the tumour cells and those of the invaded tissue. Broders [59] classified DG into three categories such that well-differentiated carcinomas (G1) represent those neoplasms in which the cancer cells and tissue organisation resemble normal cells and tissues, respectively, while poorly differentiated carcinomas (G3) correspond to those in which the tumour cells lose this similarity to conventional cells and in which the tissue has altered its normal structure (chordonal pattern). In between these two categories are moderately differentiated carcinomas (G2) with characteristics of both patterns. There are different papers that analyse the relevance of GD in the prognosis of oral cavity SCC [60,61,62]. One of the most notable is that of Chuang et al. [63] who report a research study analysing the risk of neck recurrence, based on DG, in patients with early stages of tongue SCC. They refer that cervical recurrence and DFS were significantly worse in tumours with moderate or poor DG while DOI did not reach statistical significance. In our cohort, most patients developed moderately differentiated tumours, and none of the patients in the pT1 subgroup had a poorly differentiated neoplasm. The association between tumour DG and depth of invasion divided by subgroups was statistically significant.

The main strength of this research is the homogeneity of the sample and the accuracy of the statistical analysis, and its major limitation is that it is retrospective, losing the benefits of prospective monitoring and significantly limiting the control over possible sources of bias.

## 5. Conclusions

Depth of invasion is a histological risk factor in patients diagnosed at early stages of squamous cell carcinoma of the oral cavity. The development of deeper tumours, taking into account the measures specified in the eighth edition of the AJCC, clearly worsens (*p* < 0.05) disease free survival but not overall survival. In addition, DOI is related to other risk factors such as positive lymph nodes, perineural invasion and tumour differentiation grade.

With regard to tumour staging, it has been found that DOI has a strong influence on the “T” category of the TNM classification but not on the overall staging of the disease, which tends to equalise between the seventh and eighth editions of the AJCC due to the N category.

For all of the previous considerations, it is advisable to always keep in mind the depth of invasion as one of the histological risk factors to be evaluated in order to decide on the best adjuvant treatment for patients, as well as to determine the time intervals for their evolutionary follow-up.

## Figures and Tables

**Figure 1 cancers-15-04882-f001:**
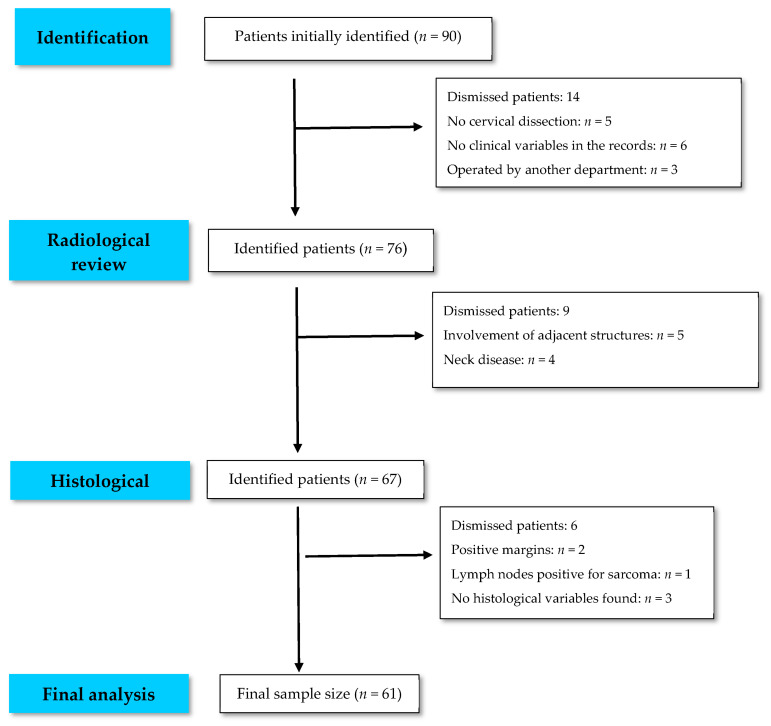
Flow chart. Final sample size.

**Figure 2 cancers-15-04882-f002:**
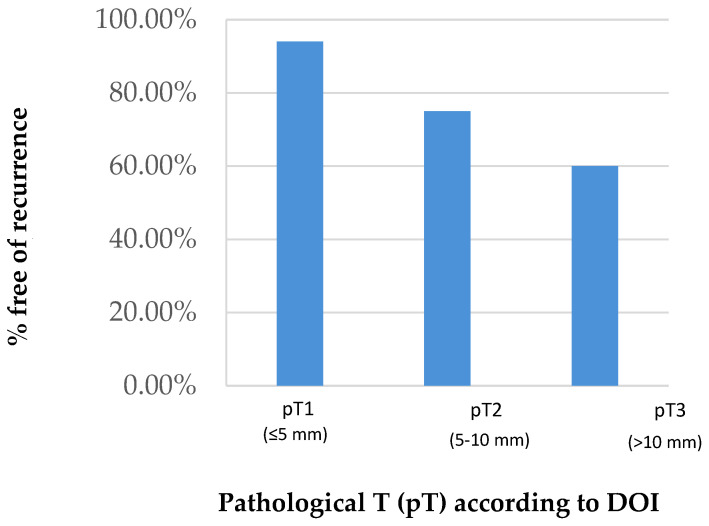
Disease-free survival according to DOI.

**Figure 3 cancers-15-04882-f003:**
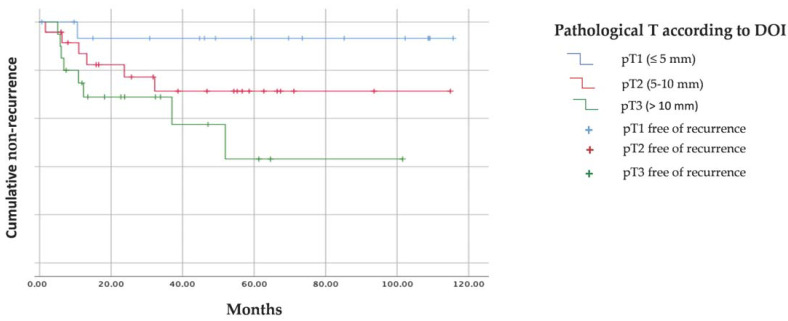
Kaplan–Meier recurrence curve for DFS according to DOI.

**Figure 4 cancers-15-04882-f004:**
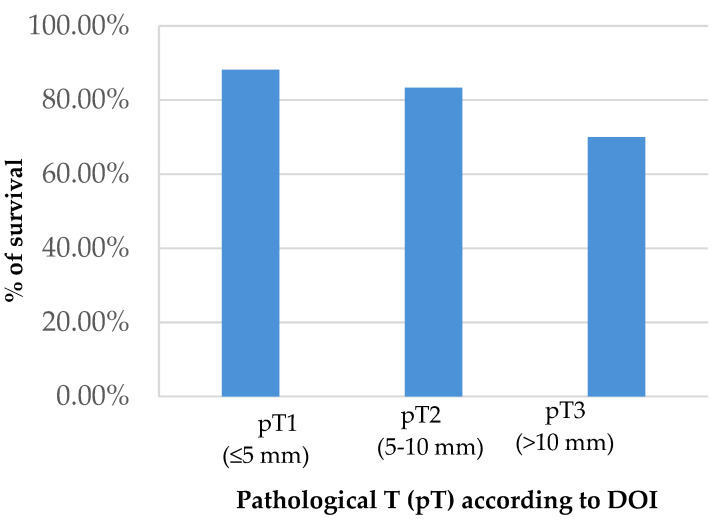
Overall survival according to DOI.

**Figure 5 cancers-15-04882-f005:**
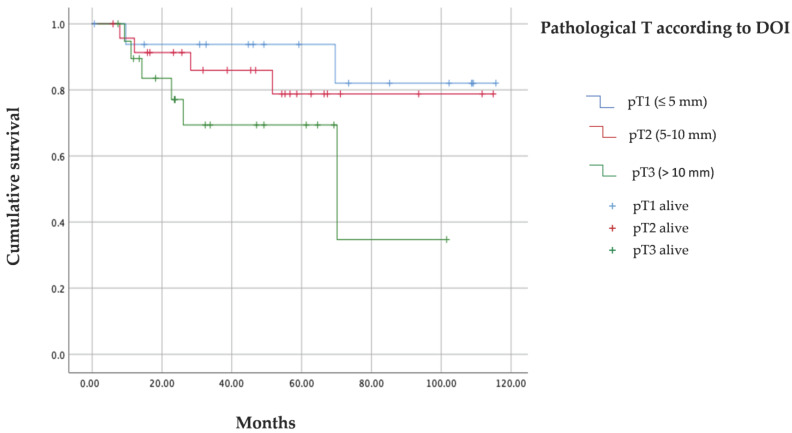
Kaplan–Meier curve for OS according to DOI.

**Figure 6 cancers-15-04882-f006:**
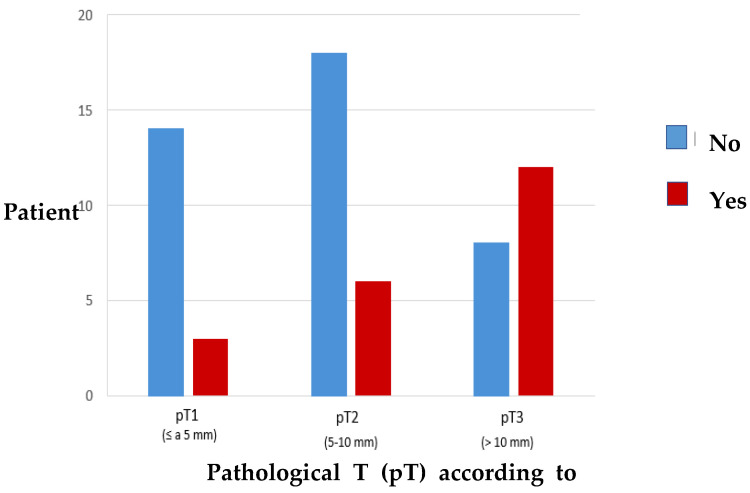
Distribution of the incidence of pathological neck nodes related to pathological T according to DOI.

**Figure 7 cancers-15-04882-f007:**
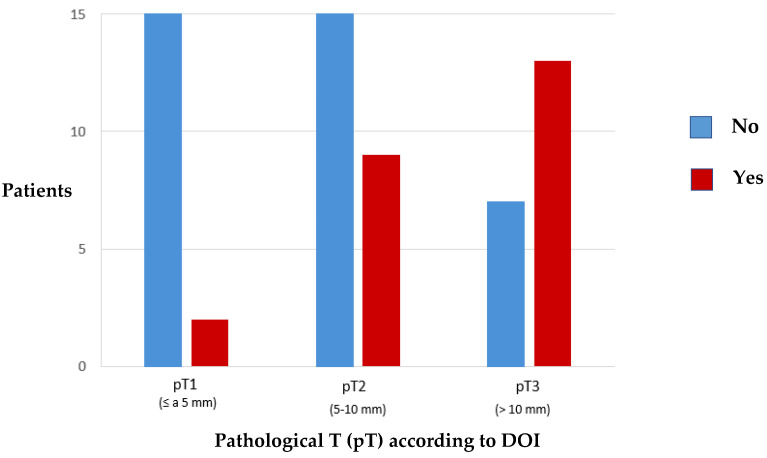
Distribution of the incidence of perineural invasion related to pathological T according to DOI.

**Figure 8 cancers-15-04882-f008:**
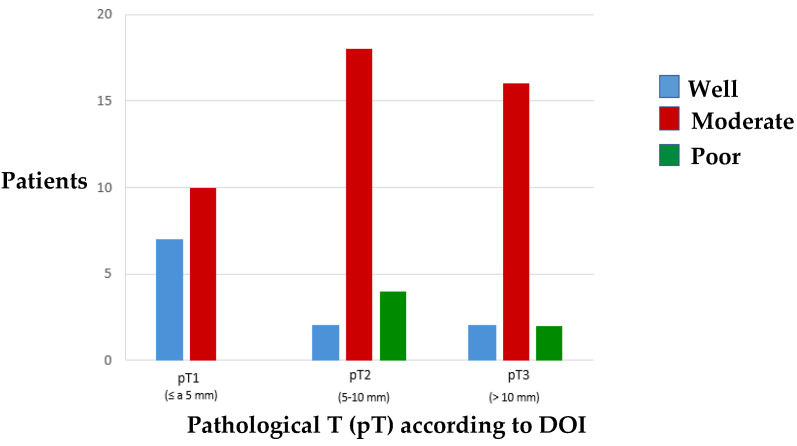
Distribution of the incidence of tumour differentiation grade related to pathological T according to DOI.

**Table 1 cancers-15-04882-t001:** Distribution of the demographic characteristics of the sample.

Variable	Category	Total Sample (*n* = 61) N (%)
* Age (years)		59.72 ± 15.74 y.o.
Age (years)	Under 40	5 (8.2%)
	40–60	25 (41.0%)
	Over 60	31 (50.8%)
Gender	Male	40 (65.6%)
	Female	21 (34.4%)
Smoking	No	35 (57.4%)
	Yes	26 (42.6%)
Alcohol intake	No	50 (82.0%)
	Yes	11 (18.0%)

* Variable age is presented as mean ± standard deviation of the mean.

**Table 2 cancers-15-04882-t002:** Distribution of the clinical characteristics of the sample.

Variable	Category	Total Sample (*n* = 61) N (%)
Clinical tumour size (cm)	T1 (under 2 cm)	27 (44.3%)
	T2 (between 2 and 4 cm)	34 (55.7%)
Tumour location	Lateral border	50 (82%)
	Dorsal tongue (submucosal)	2 (3.3%)
	Ventral tongue	9 (14.7%)
CT scan	No	1 (1.6%)
	Yes	60 (98.4%)
MR imaging	No	47 (77.0%)
	Yes	14 (23.0%)
Radiological tumour size (cm)	Not appreciable	24 (39.3%)
	T1 (under 2 cm)	23 (37.7%)
	T2 (between 2 and 4 cm)	14 (23%)
Reconstruction	Direct closure, granulation	28 (45.9%)
	Local flap	15 (24.6%)
	Microsurgery	18 (29.5%)
Elective neck dissection	Unilateral	33 (54.1%)
	Bilateral	28 (45.9%)
Postoperative radiotherapy	No	32 (52.5%)
	Yes	29 (47.5%)
Postoperative chemotherapy	No	59 (96.7%)
	Yes	2 (3.3%)

**Table 3 cancers-15-04882-t003:** Distribution of the histological characteristics of the sample.

Variable	Category	Total Sample (*n* = 61) N (%)
* Depth of invasion (mm)		9.03 ± 5.34 mm
Tumour differentiation grade	Well	11 (18.0%)
	Moderate	44 (72.2%)
	Poor	6 (9.8%)
Perineural invasion	No	37 (60.7%)
	Yes	24 (39.3%)
Lymphovascular invasion	No	56 (91.8%)
	Yes	5 (8.2%)
Pathological T according to DOI (mm)	pT1 (under 5 mm)	17 (27.9%)
	pT2 (between 5 and 10 mm)	24 (39.43%)
	pT3 (over 10 mm)	20 (32.8%)
Pathological T according to size (cm)	pT1 (under 2 cm)	30 (49.2%)
	pT2 (between 2 and 4 cm)	28 (45. 9%)
	pT3 (over 4 cm)	3 (4.9%)
Pathological T of the 7th edition of the AJCC	pT1	30 (49.2%)
	pT2	28 (45.9%)
	pT3	3 (4.9%)
Pathological T of the 8th edition of the AJCC	pT1	15 (24.6%)
	pT2	25 (41.0%)
	pT3	21 (34.4%)
Histological positive neck	No	40 (65.6%)
	Yes	21 (34.4%)
Histological neck	pN0	40 (65.5%)
	pN+ ipsilaterally	17 (27.9%)
	pN+ bilaterally	4 (6.6%)
Extranodal extension	No	59 (95.1%)
	Yes	3 (4.9%)
Histological neck according to the 7th edition of the AJCC	pN0	40 (65.6%)
	pN1	10 (16.4%)
	pN2b	7 (11.5%)
	pN2c	4 (6.5%)
Histological neck according to the 8th edition of the AJCC	pN0	40 (65.6%)
	pN1	9 (14.8%)
	pN2a	1 (1.6%)
	pN2b	6 (9.8%)
	pN2c	3 (4.9%)
	pN3b	2 (3.3%)
Staging according to the 7th edition of the AJCC	Stage I	24 (39.3%)
	Stage II	15 (24.7%)
	Stage III	11 (18.0%)
	Stage IVa	11 (18.0%)
Staging according to the 8th edition of the AJCC	Stage I	13 (21.3%)
	Stage II	19 (31.1%)
	Stage III	17 (27.9%)
	Stage IVa	10 (16.4%)
	Stage IVb	2 (3.3%)

* Variable depth of tumour invasion is presented as mean ± standard deviation of the mean.

**Table 4 cancers-15-04882-t004:** Distribution of evolutive control characteristics of the sample.

Variable	Category	Total Sample (*n* = 61) N (%)
Loco-regional recurrence	No	47 (77.0%)
	Yes	14 (23.0%)
Distant recurrence	No	56 (91.8%)
	Yes	5 (8.2%)
Total recurrence	No	47 (77.0%)
	Yes	14 (23.0%)
Deaths	No	49 (80.3%)
	Yes	12 (19.7%)

**Table 5 cancers-15-04882-t005:** Distribution of demographic characteristics depending on pathologic T according to DOI.

Variable	Category	Subgroup pT1 % (CI 95)	Subgroup pT2 % (CI 95)	Subgroup T3 % (CI 95)	*p* Value
* Age (years old)		63.47 ± 17.80	58.04 ± 15.32	58.55 ± 14.65	0.411
Age (years old)	Under 40	40.0% (9.4–79.1)	60.0% (20.9–90.6)	0.0%	0.357
	40–60	20.0% (8.1–38.4)	36.0% (19.5–55.5)	44.0% (26.1–63.2)	
	Over 60	32.3% (17.9–49.7)	38.7% (23.2–56.2)	29.0% (15.4–46.3)	
Gender	Male	35.0% (21.7–50.4)	35.0% (21.7–50.4)	30.0% (17.6–45.2)	0.228
	Female	14.3% (4.2–33.4)	47.6% (27.7–68.1)	38.1% (19.9–59.3)	
Smoking	No	22.9% (11.4–38.5)	40.0% (25.1–56.5)	37.1% (22.7–52.7)	0.542
	Yes	34.6% (18.7–53.7)	38.5% (21.8–57.6)	26.9% (12.9–45.7)	
Alcohol intake	No	28.0% (17.0–41.4)	38.0% (25.5–51.8)	34.0% (22.1–47.7)	0.881
	Yes	27.3% (8.3–56.5)	45.4% (20.0–73.0)	27.3% (8.3–56.5)	

*: Variable age is presented as mean ± standard deviation of the mean.

**Table 6 cancers-15-04882-t006:** Distribution of clinical characteristics depending on pathologic T according to DOI.

Variable	Category	Subgroup pT1 % (CI 95)	Subgroup pT2 % (CI 95)	Subgroup pT3 % (CI 95)	*p* Value
Clinical tumour size (cm)	T1 (under 2 cm)	48.1% (30.3–66.4)	37.1% (20.9–55.8)	14.8% (5.2–31.5)	0.002
	T2 (between 2 and 4 cm)	11.8% (4.1–25.6)	41.2% (25.9–57.9)	47.0% (31.1–63.5)	
Tumour location	Lateral border	30.0% (18.7–43.6)	36.0% (23.8–49.8)	34.0% (22.1–47.7)	0.726
	Dorsal tongue (submucosal)	0.0%	50.0% (6.1–93.9)	50.0% (6.1–93.9)	
	Ventral tongue	22.2% (4.9–54.4)	55.6% (25.4–82.7)	22.2% (4.9–54.4)	
CT scan	No	0.0%	0.0%	100.0%	0.353
	Yes	28.3% (18.1–40.6)	40.0% (28.3–52.6)	31.7% (21–44.1)	
MR imaging	No	27.7% (16.5–41.5)	40.4% (27.3–54.7)	31.9% (20.0–46.0)	0.946
	Yes	28.6% (10.5–54.5)	35.7% (15.1–65.5)	35.7% (15.1–65.5)	
Radiological tumour size (cm)	Not appreciable	33.3% (17.2–53.2)	45.8% (27.3–65.3)	20.9% (8.4–39.8)	0.048
	T1 (under 2 cm)	39.1% (21.4–59.4)	30.4% (14.8–50.7)	30.5% (14.8–50.7)	
	T2 (between 2 and 4 cm)	0.0%	42.9% (20.3–68.1)	57.1% (31.9–79.7)	
Reconstruction	Direct closure, granulation	50.0% (32.2–67.8)	50.0% (32.2–67.8)	0.0%	<0.001
	Loco-regional flap	6.6% (0.7–27.2)	46.7% (23.9–70.6)	46.7% (23.9–70.6)	
	Microsurgery	11.1% (2.4–31.1)	16.7% (4.9–38.1)	72.2% (49.4–88.5)	
Elective neck dissection	Unilateral	39.4% (22.4–54.8)	51.5% (36.2–69.5)	9.1% (2.7–23)	<0.001
	Bilateral	14.3% (5.0–30.5)	25.0% (11.9–42.9)	60.7% (42.3–77)	
Postoperative radiotherapy	No	43.8% (27.7–60.9)	50.0% (33.3–66.7)	6.2% (1.3–18.6)	<0.001
	Yes	10.3% (3.0–25.1)	27.6% (14.0–45.4)	62.1% (44–77.9)	
Postoperative chemotherapy	No	28.8% (18.5–41.2)	39.0% (27.3–51.7)	32.2% (21.4–44.8)	0.663
	Yes	0.0%	50.0% (6.1–93.9)	50.0% (6.1–93.9)	

**Table 7 cancers-15-04882-t007:** Distribution of histological characteristics depending on pathologic T according to DOI.

Variable	Category	Subgroup pT1 % (CI 95)	Subgroup pT2 % (CI 95)	Subgroup pT3 % (CI 95)	*p* Value
* DOI (mm)		3.41 ± 1.32	7.88 ± 1.96	15.20 ± 3.73	<0.001
Tumour differentiation grade	Well	63.6% (34.8–86.3)	18.2% (4.0–46.7)	18.2% (4.0–46.7)	0.034
	Moderate	22.7% (12.3–36.6)	40.9% (27.3–55.6)	36.4% (23.4–51.1)	
	Poor	0.0%	66.7% (28.6–92.3)	33.3% (7.7–71.4)	
Perineural invasion	No	40.5% (25.9–96.6)	40.5% (25.9–96.6)	19.0% (8.9–33.6)	0.004
	Yes	8.3% (1.8–24.1)	37.5% (20.4–57.4)	54.2% (34.7–72.7)	
Lymphovascular invasion	No	30.4% (19.5–43.2)	37.5% (25.7–50.5)	32.1% (21.0–45.0)	0.334
	Yes	0.0%	60.0% (20.9–90.6)	40.0% (9.4–79.1)	
Pathological T according to size (cm)	pT1 (under 2 cm)	50.0% (32.8–67.2)	40.0% (24–57.8)	10.0% (2.9–24.3)	0.001
	pT2 (between 2 and 4 cm)	7.1% (1.5–21.0)	39.3% (23.0–57.7)	53.6% (35.5–70.9)	
	pT3 (over 4 cm)	0.0%	33.3% (3.9–82.3)	66.7% (17.7–96.1)	
Pathological T of the 7th edition of the AJCC	pT1	50.0% (32.8–67.2)	40.0% (24.0–57.8)	10.0% (2.9–24.3)	0.001
	pT2	7.1% (1.5–21.0)	39.3% (23.0–57.7)	53.6% (35.5–70.9)	
	pT3	0.0%	33.3% (3.9–82.3)	66.7% (17.7–96.1)	
Pathological T of the 8th edition of the AJCC	pT1	100.0%	0.0%	0.0%	<0.001
	pT2	8.0% (1.7–23.0)	92.0% (76.7–98.3)	0.0%	
	pT3	0.0%	4.8% (0.5–20.2)	95.2% (79.8–99.5)	
Histological positive neck	No	35.0% (21.7–50.4)	45.0% (30.4–60.3)	20.0% (9.9–30.2)	0.012
	Yes	14.3% (4.2–33.4)	28.6% (12.9–49.7)	57.1% (36.2–76.3)	
Histological neck	pN0	35.0% (21.7–50.4)	45.0% (30.4–60.3)	20.0% (9.9–34.2)	0.045
	pN+ ipsilateral	17.6% (5.2–40.0)	29.4% (12.2–53.0)	53.0% (30.3–74.6)	
	pN+ bilateral	0.0%	25.0% (2.8–71.6)	75.0% (28.4–97.2)	
Extranodal extension	No	29.3% (18.8–41.8)	37.9% (26.3–50.8)	32.8% (21.7–45.4)	0.478
	Yes	0.0%	66.7% (17.7–96.1)	33.3% (3.9–82.3)	
Histological neck according to the 7th edition of the AJCC	pN0	35.0% (21.7–50.4)	45.0% (30.4–60.3)	20.0% (9.9–34.2)	0.099
	pN1	20.0% (4.4–50.3)	20.0% (4.4–50.3)	60.0% (30.4–84.7)	
	pN2b	14.2% (1.6–50.1)	42.9% (13.9–76.5)	42.9% (13.9–76.5)	
	pN2c	0.0%	25.0% (2.8–71.6)	75.0% (28.4–97.2)	
Histological neck according to the 8th edition of the AJCC	pN0	35.0% (21.7–50.4)	45.0% (30.4–60.3)	20.0% (9.9–34.2)	0.091
	pN1	22.2% (4.9–54.4)	22.2% (4.9–54.4)	55.6% (25.4–82.7)	
	pN2a	0.0%	0.0%	100%	
	pN2b	16.7% (1.9–55.8)	33.3% (7.7–71.4)	50.0% (16.7–83.3)	
	pN2c	0.0%	0.0%	100.0%	
	pN3b	0.0%	100.0%	0.0%	
Staging according to the 7th edition of the AJCC	Stage I	52.0% (33.1–70.5)	36.0% (19.5–55.5)	12.0% (3.5–28.7)	0.001
	Stage II	6.6% (0.7–27.2)	66.7% (41.6–86.0)	26.7% (9.7–51.7)	
	Stage III	18.2% (4.0–46.7)	18.2% (4.0–46.7)	63.6% (34.8–86.3)	
	Stage Iva	9.1% (1.1–38.1)	36.3% (9.3–60.6)	54.6% (30.4–84.7)	
Staging according to the 8th edition of the AJCC	Stage I	100.0%	0.0%	0.0%	<0.001
	Stage II	5.3% (0.6–22.1)	94.7% (77.9–99.4)	0.0%	
	Stage III	11.8% (2.5–32.7)	11.8% (2.5–32.7)	76.4% (53.3–91.5)	
	Stage Iva	10.0% (1.2–41.4)	20.0% (4.9–54.4)	70.0% (34.8–89.6)	
	Stage Ivb	0.0%	100.0%	0.0%	

* Variable depth of invasion is presented as mean ± standard deviation of the mean.

**Table 8 cancers-15-04882-t008:** Distribution of evolutive control characteristics depending on pathologic T according to DOI.

Variable	Category	Subgroup pT1 % (CI 95)	Subgroup pT2 % (CI 95)	Subgroup pT3 % (CI 95)	*p* Value
Loco-regional recurrence	No	34.8% (22.3–49.1%)	39.1% (26.0–53.5%)	26.1% (15.1–40.0%)	0.056
	Yes	6.7% (0.7–27.2%)	40.0% (18.8–64.7%)	53.3% (29.4–76.1%)	
Distant recurrence	No	28.6% (18.0–41.3%)	39.3% (27.3–52.4%)	32.1% (21.0–45.0%)	0.901
	Yes	20.0% (2.3–62.9%)	40.0% (9.4–79.1%)	40.0% (9.4–79.1%)	
Total recurrence	No	34.8% (22.3–49.1%)	39.1% (26–53.5%)	26.1% (15.1–40.0%)	0.056
	Yes	6.7% (0.7–27.2%)	40% (18.8–64.7%)	53.3% (29.4–76.1%)	
Deaths	No	30.6% (19.1–44.3%)	40.8% (27.9–54.8%)	28.6% (24.3–75.7%)	0.340
	Yes	16.7% (3.6–43.6%)	33.3% (12.5–61.2%)	50.0% (24.3–75.7%)	

**Table 9 cancers-15-04882-t009:** Estimated disease-free survival for pT1, pT2 and pT3.

Pathological T According to T	Media (Estimated ± Deviation)	Log-Rank (Mantel–Cox) (*p* Value)
pT1 (≤5 mm)	108.67 ± 6.77 months	0.043
pT2 (5–10 mm)	86.44 ± 9.92 months	
pT3 (>10 mm)	57.93 ± 11.26 months	

**Table 10 cancers-15-04882-t010:** Estimated overall survival in months for pT1, pT2 and pT3.

Pathological T According to DOI	Media (Estimated ± Deviation)	Log-Rank (Mantel–Cox) (*p* Value)
pT1 (≤5 mm)	103.64 ± 7.89 months	0.139
pT2 (5–10 mm)	96.59 ± 8.24 months	
pT3 (>10 mm)	64.96 ± 11.03 months	

**Table 11 cancers-15-04882-t011:** Modification of the T category in relation to the TNM classification of the eighth edition of the AJCC.

	Global T Category (Size and DOI) According to the 8th Edition of the AJCC			
	Patients pt1	Patients pt2	Patients pt3	Total
Patients clinical T1	13	10 *	4 *	27
Patients clinical T2	4	14	16 *	34
Total	17	24	20	61

* In bold, patients who upgraded their T category.

## Data Availability

The data presented in this study are available on request from the corresponding author. The data are not publicly available, due to data-protection regulations.

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
