# Peer review of "Depth of Invasion: Influence of the Latest TNM Classification on the Prognosis of Clinical Early Stages of Oral Tongue Squamous Cell Carcinoma and Its Association with Other Histological Risk Factors"

_cancers, 2023, doi:10.3390/cancers15194882_

Round 1

Reviewer 1 Report

The Authors conducted a very original study regarding the prognostic indices most involved in the survival of patients with oral  cancer. The paper is well written in English and presents innovative features for the research carried out. It does not require revisions. 

Author Response

I have made changes to the article being evaluated for publication in Cancers.

Reviewer 1 gave the "go-ahead" to the submitted article so I have focused on the suggested changes from reviewers 2 and 3.

Thank you very much and I look forward to your response.

Reviewer 2 Report

The authors aimed to assess the prognosis, according to depth of invasion, of patients with clinical early stages of oral tongue squamous cell carcinoma, to evaluate the influence of depth of invasion in the latest TNM classification as well as in the global staging system and to analyse its relation to other histological risk factors.

The study covers some issues that have been overlooked in other similar topics. The structure of the manuscript appears adequate and well divided in the sections. Moreover, the study is easy to follow, but some issues should be improved. Some of the comments that would improve the overall quality of the study are:

a. Authors must pay attention to the technical terms acronyms they used in the text.

b. English language needs to be revised.

c. Conclusion Section: This paragraph required a general revision to eliminate redundant sentences and to add some "take-home message".

Minor editing of English language required

Author Response

Review report 2:

The study covers some issues that have been overlooked in other similar topics. The structure of the manuscript appears adequate and well divided in the sections. Moreover, the study is easy to follow, but some issues should be improved. Some of the comments that would improve the overall quality of the study are:

Thank you very much for your answer and time spent reading the paper. I outline the changes made in relation to your comments.

  1. Authors must pay attention to the technical terms acronyms they used in the text.

Line 226: postoperative radiotherapy (PORT) included.

Line 98: PNI inserted.

Line 234:PNI inserted.

Line 344: PNI inserted.

Line 479: PNI inserted.

  1. English language needs to be revised.

Line 36: “assess” instead of “asses”.

Line 53: “it´s” en vez de “its”.

Line 84: “can modify” instead of “are capable of modifying”.

Line 107: “were” instead of “are”

Line 115: “tongue” instead of “tonge”.

Line 140: “survival,” instead of survival.

Line 219: “regarding” instead of “with regard to”

Line 235: “neck,” instead of “neck”

Line 324: “most” instead of “vast majority of”

Line 344: PNI instead of “perineural invasion”

Line 395: “the” instead of “it is clear that the”.

Line 413: “can modify” instead of “is capable of modifying”.

Line 421: “and instead of “and also”.

Line 513: “most” instead of “the majority of”.

  1. Conclusion Section: This paragraph required a general revision to eliminate redundant sentences and to add some "take-home message".

A new paragraph has been included (line 531-544). Thank you for your comment.

Reviewer 3 Report

Line 40 is in present and it must be in past (were). Same situation in Line 106.

Line 41 and 42. Need to adjust sentence punctuation

Line 50. Multivariate analysis showed association between ???? with perineural invasion.

Some terms were not found in MeSH (Medical Subject Headings)

In the  Introduction, references in lines from 69 to 82 are in parentheses

In Introduction, the  lines 98 to 100 contents do not have a link to the text in the previous paragraph because it has an affirmative construction and does not show a research question.

Review formatting of lines 111 and 112.

Include “n=61” in the Final sample size of Figure 1. Flow chart. Final sample size.

Line 145 and 146 mention the limitations of the research, so this is content for the Discussion topic.

Line 240, specify IPN, as well as check the insertion of the specification/name in text before the abbreviations. Perineural invasion (PNI) information was specified in Line 477, but PNI was mentioned in Line 240 by the first time.

Topic Results is very detailed.

Discussion presents an adequate structure and conclusion compatible with the objective of the manuscript and the obtained result.

No explanation for the presence of “(92 thesis)” in Line 508.

I suggest including a final paragraph in the Discussion Topic with the limitations and strengths of the research associated with the manuscript.

Quality of the English Language is satisfactory.

Author Response

Review report 3:

Thank you very much for your answer and time spent reading the paper. I outline the changes made in relation to your comments.

Comments and Suggestions for Authors.

Line 40 is in present, and it must be in past (were). it has been revised.Same situation in Line 106. it has beenrevised.

Line 41 and 42. Need to adjust sentence punctuation. it has been revised.

Line 50. Multivariate analysis showed association between depth of invasion andperineural invasion.

Some terms were not found in MeSH (Medical Subject Headings): keywords have been reviewed and changed. Thank you for your comment.

In the Introduction, references in lines from 69 to 82 are in parentheses: it has been revised.

In Introduction, the lines 98 to 100 contents do not have a link to the text in the previous paragraph because it has an affirmative construction and does not show a research question. The paragraph has been deleted.

Review formatting of lines 111 and 112. It has been revised.

Include “n=61” in the Final sample size of Figure 1. Flow chart. Final sample size. It has been included.

Line 145 and 146 mention the limitations of the research, so this is content for the Discussion topic. it has been deleted from methods and included at the end of the Discussion (lines 518-521)

Line 240, specify IPN, as well as check the insertion of the specification/name in text before the abbreviations. Perineural invasion (PNI) information was specified in Line 477, but PNI was mentioned in Line 240 by the first time. Thank you for your comment. It has been revised. PNI inserted in line 98, 234, 275, 344, 479,

Topic Results is very detailed. Thank you for your comment.

Discussion presents an adequate structure and conclusion compatible with the objective of the manuscript and the obtained result. Thank you for your comment.

No explanation for the presence of “(92 thesis)” in Line 508. it has been revised. [63] instead of 92 tesis

I suggest including a final paragraph in the Discussion Topic with the limitations and strengths of the research associated with the manuscript.Included at the end of the Discussion (lines 518-521).

English language needs to be revised.

Line 36: “assess” instead of “asses”.

Line 53: “it´s” en vez de “its”.

Line 84: “can modify” instead of “are capable of modifying”.

Line 107: “were” instead of “are”

Line 115: “tongue” instead of “tonge”.

Line 140: “survival,” instead of survival.

Line 219: “regarding” instead of “with regard to”

Line 235: “neck,” instead of “neck”

Line 324: “most” instead of “vast majority of”

Line 344: PNI instead of “perineural invasion”

Line 395: “the” instead of “it is clear that the”.

Line 413: “can modify” instead of “is capable of modifying”.

Line 421: “and instead of “and also”.

Line 513: “most” instead of “the majority of”.
